# Vegetation Dynamic Changes and Their Response to Ecological Engineering in the Sanjiangyuan Region of China

**Xiaohui Zhai [1,2], Xiaolei Liang [3,*,†], Changzhen Yan [1,4,†], Xuegang Xing [1,2], Haowei Jia [1,2], Xiaoxu Wei [5] and Kun Feng [1,2]**

1. Key Laboratory of Desert and Desertification, Northwest Institute of Eco-Environment and Resources, Chinese Academy of Sciences, Lanzhou 730000, China; zhaixiaohui@lzb.ac.cn (X.Z.); yancz@lzb.ac.cn (C.Y.); xingxuegang@lzb.ac.cn (X.X.); haoweijia@lzb.ac.cn (H.J.); fengkun@lzb.ac.cn (K.F.)
2. University of Chinese Academy of Sciences, Beijing 100049, China
3. Department of Management, Taiyuan Normal University, Jinzhong 030619, China
4. National Earth System Science Data Center, Beijing 100020, China
5. College of Social Development and Public Administration, Northwest Normal University, Lanzhou 730070, China; weixiaoxu@lzb.ac.cn
* Correspondence: liangxl@lzb.ac.cn
† These authors have equal contribution.

**Abstract:** In recent decades, the vegetation of the Sanjiangyuan region has undergone a series of changes under the influence of climate change, and ecological restoration projects have been implemented. In this paper, we analyze the spatiotemporal dynamics of vegetation in this region using the satellite-retrieved normalized difference vegetation index (NDVI) from the global inventory modeling and mapping studies (GIMMS) and moderate resolution imaging and spectroradiometer (MODIS) datasets during the past 34 years. Specifically, the characteristics of vegetation changes were analyzed according to the stage of implementation of different ecological engineering programs. The results are as follows. (1) The vegetation in 65.6% of the study area exhibited an upward trend, and in 53.0% of the area, it displayed a large increase, which was mainly distributed in the eastern part of the study area. (2) The vegetation NDVI increased to differing degrees during stages of ecological engineering. (3) The NDVI in the western part of the Sanjiangyuan region is mainly affected by temperature, while in the northeastern part, the NDVI is affected more by precipitation. In the southern part, however, vegetation growth is affected neither by temperature nor by precipitation. On the whole region, vegetation growing is more affected by temperature than by precipitation. (4) The impacts of human activities on vegetation change are both positive and negative. In recent years, ecological engineering projects have had a positive impact on vegetation growth. This study can help us to correctly understand the impact of climate change on vegetation growth, so as to provide a scientific basis for the evaluation of regional ecological engineering effectiveness and the formulation of ecological protection policies.

**Keywords:** Sanjiangyuan region; NDVI; ecological protection and construction projects; climate factor

## 1. Introduction

Vegetation is the main body of the terrestrial ecosystem, connecting the material circulation and energy flow of soil, hydrosphere and atmosphere, and playing an important role in regulating terrestrial carbon balance and climate system [1,2]. Monitoring the dynamic changes of vegetation has important scientific significance and practical value. Vegetation cover changes are strongly influenced

by climate change and human activities [3,4], and the response of vegetation to external disturbances has become a hot topic in the academic circles in the domestic and overseas [5–7].

Located in the hinterland of the Qinghai-Tibet Plateau, the Sanjiangyuan region is the birthplace of the Yangtze River, the Yellow River and the Lantsang River in China. It is a critical area for water resource conservation, and its ecosystem is very sensitive and fragile [8,9]. The typical alpine vegetation system of Sanjiangyuan plays an important role in the study of global climate change and response [10]. Additionally, this area has become a hot spot for studying vegetation changes [11,12]. In past decades, the vegetation ecosystem in this area has degraded significantly due to the dual impact of climate change and human activities, manifested as grassland degradation, land desertification, agricultural and animal husbandry output decline, and so on [13].

Since the 1980s, China has launched a series of ecological restoration programs to mitigate these increasingly devastating environmental problems. These projects include the Protection Forest System project in the middle and upper reaches of the Yangtze River (PFSYR) and the First and Second Phase Plans on Ecological Protection and Construction in Qinghai Sanjiangyuan (EPCQS I and II). PFSYR is one of the eight major ecological projects in the world. The first phase of the PFSYR project, which was implemented from 1989 to 2000, aimed to conserve water and soil and realize the virtuous cycle of natural ecology in the Yangtze River Basin [14]. EPCQS is an ecological construction project launched after the establishment of the Sanjiangyuan National Nature Reserve. The first phase of the project, with an investment of 7.5 billion yuan and an implementation period from 2005 to 2012, included measures such as ecological migration, livestock reduction projects, returning farmland to forests and grasslands and desertification prevention and control, and it has achieved remarkable ecological restoration results [15]. The second phase of the project with an investment of 16 billion yuan was launched in 2013.

Remote sensing data have been widely used in monitoring vegetation dynamics and ecological restoration in large areas due to their wide coverage area and high time-phase resolution [16,17]. In recent years, many scholars have used remote sensing technology to study the dynamic changes in vegetation and its driving factors in Sanjiangyuan. Initially, scholars used GIMMS NDVI (normalized difference vegetation index from the Global Inventory Modeling and Mapping Studies) data to monitor for about 20 years [12,18], which was started in 1982 and had a spatial resolution of 8 km. Around 2000, higher resolution vegetation remote sensing data sources appeared, such as SPOT-VEGETATION data with a resolution of 1 km and moderate resolution imaging and spectroradiometer (MODIS) NDVI (normalized difference vegetation index from the moderate-resolution imaging and spectroradiometer) data with a resolution of 1 km, 500 m and 250 m. Most of the studies after 2000 were based on these two datasets [10,11,16,19–21]. However, these studies mostly used a single remote sensing data source, and their shortcomings are obvious—problems of low resolution and short research period cannot be solved at the same time. Meanwhile, there is a lack of analysis on the trend of vegetation change in recent years, in which policy making and ecological restoration are more important, and the effects of ecological projects on vegetation restoration have not been mentioned.

In this study, we investigated the spatiotemporal variation characteristics of the vegetation in the Sanjiangyuan region over the past 34 years by using the GIMMS NDVI and MODIS NDVI datasets, supplemented by linear regression models, change tendency analysis and related analysis methods. Specifically, the characteristics of changes in vegetation were analyzed according to the stage of implementation of different ecological engineering programs. Furthermore, we combined grid meteorological data to discuss the main climatic factors that affect vegetation changes and finally discussed the impact of human activities based on statistical data. These can not only correctly help to understand the impact of climate change on the growth of the vegetation, but also evaluate the effect of the implementation of ecological restoration policies, and provide basic data for the implementation of further ecological restoration projects. The research group believed that if the relationship between the two types of datasets was established through correlation analysis, the unicity of vegetation research data in this area could be broken and the research period could be effectively extended.

## 2. Study Area

The Sanjiangyuan region is located in the Qinghai–Tibet Plateau and covers an area of $27.8 \times 10^4$ km$^2$ (Figure 1), with an altitude range of 2672–6560 m. This region has a typical continental plateau climate, with strong solar radiation and long sunshine durations [22]. The average annual temperature is −2.05 °C, and the average precipitation is 383.39 mm. Both of these gradually decrease from southeast to northwest. Furthermore, there are more than 80 rivers in these three river systems, including the Tuotuo, Chumal, Buqu, Dangqu, Duoqu and Requ Rivers. Sanjiangyuan is a multilake area that has more than 1800 lakes, including 188 natural lakes with an area of more than 0.5 km$^2$. This area consists of 15 counties and one town (Tanggula Mountain Town under the jurisdiction of Golmud City).

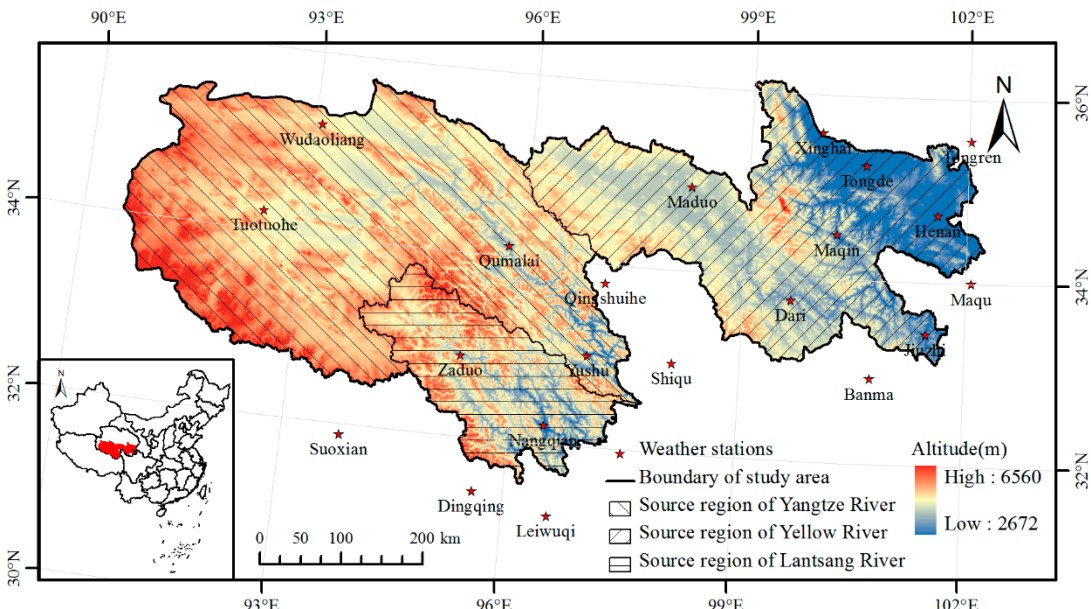

**Figure 1.** Location of the Sanjiangyuan region and distribution of weather stations.

## 3. Data and Methods

### 3.1. Data Sources and Preprocessing

3.1.1. NDVI Data and Preprocessing

The normalized difference vegetation index (NDVI) is the most commonly used index of vegetation growth status [23]. The remote sensing data used in this study were NDVI data products of NOAA/AVHRR and TERRA/AQUA MODIS. The NOAA/AVHRR NDVI datasets use the latest version of GIMMS (Global Inventory Modeling and Mapping Studies) NDVI3g_v1, with a time span from July 1981 to December 2015, a temporal resolution of 15 d, and a spatial resolution of 8 km. These NDVI datasets are the longest time series data at present [24]. Compared with other NDVI data, GIMMS NDVI data has smaller errors and higher accuracy and has been widely used in the study of global and regional large-scale vegetation changes [25]. The MODIS NDVI data uses the MOD13A2 products freely provided by NASA, with a time span from 2000 to 2015, a temporal resolution of 16 d and a spatial resolution of 1 km.

As mentioned in the introduction, most of the studies adopt a single data source, which cannot solve the problems of low resolution and short research period at the same time. Therefore, we wanted to combine low-resolution, long-time-series GIMMS NDVI data and high-resolution, short-time-series MODIS NDVI data to solve that. Existing studies believe that it is possible to combine GIMMS and MODIS data to study vegetation changes in the Northern Hemisphere [26]. Some scholars also have tried to combine them for their research [27–29]. In addition, considering the great difference in vegetation growth in different months, we made a correlation analysis of the NDVI data in each month. Then we used the

correlation coefficient to extend the MODIS NDVI data. These minimized the error in the calculation as much as possible. First, we converted the projection of the two datasets to the Albers projection. Second, we resampled the GIMMS NDVI data from 8 to 1 km to make the two datasets consistent in spatial resolution. Third, we used the MVC (maximum value composite) [30] to obtain the monthly NDVI value so that each pixel is the maximum NDVI value of the month. This method can further eliminate the influence of clouds, atmosphere, solar altitude angle, etc. (Equation (1)).

$$NDVI_i = Max(NDVI_{ij}) \tag{1}$$

where $NDVI_i$ is the NDVI value of the *i*-th month, and $NDVI_{ij}$ is the NDVI value of the *j*-th ten-day of the *i*-th month. Finally, the grid values with minimum values >0 per month were selected to calculate the linear correlation coefficient (Table 1) between the GIMMS NDVI and the MODIS NDVI. Based on this relationship, the time series of the MODIS NDVI was extended by the GIMMS monthly NDVI from 1982 to 2000.

**Table 1.** The correlation between the Global Inventory Modeling and Mapping Studies (GIMMS) normalized difference vegetation index (NDVI) and the moderate resolution imaging and spectroradiometer (MODIS) NDVI.

| Month | *n* | a | b | $R^2$ | RMSE |
|-------|------|-------|--------|-------|-------|
| 1 | 275,927 | 0.866 | 0.005 | 0.628 | 0.039 |
| 2 | 275,808 | 0.893 | 0.008 | 0.661 | 0.034 |
| 3 | 275,810 | 0.891 | 0.005 | 0.658 | 0.036 |
| 4 | 275,691 | 0.977 | −0.001 | 0.707 | 0.035 |
| 5 | 275,578 | 0.862 | −0.029 | 0.755 | 0.084 |
| 6 | 276,054 | 0.971 | −0.007 | 0.811 | 0.077 |
| 7 | 276,686 | 0.988 | 0.010 | 0.800 | 0.092 |
| 8 | 276,775 | 0.954 | 0.020 | 0.784 | 0.094 |
| 9 | 276,262 | 0.902 | 0.003 | 0.789 | 0.089 |
| 10 | 275,819 | 0.925 | −0.012 | 0.714 | 0.070 |
| 11 | 275,635 | 0.900 | 0.000 | 0.658 | 0.044 |
| 12 | 274,984 | 0.873 | 0.005 | 0.619 | 0.041 |

### 3.1.2. Meteorological Data Sources and Preprocessing

The meteorological data were derived from the China Meteorological Science Data Sharing Service Network (http://cdc.cma.gov.cn/home.do), which contains the daily mean temperature and precipitation data from 21 meteorological stations in the Sanjiangyuan region and its surrounding areas from 1982 to 2015.

As the Sanjiangyuan area is located in the hinterland of the Qinghai–Tibet Plateau, its terrain is undulating, and its meteorological stations are sparse, commonly used interpolation methods such as kriging and inverse distance weighting (IDW) methods have difficulty achieving high accuracy in this area [31]. Therefore, we used the ANUSPLIN software package version 4.3 [32], which implements the thin-plate smoothing spline procedure described by Hutchinson [33]. This method can introduce multiple impact factors (such as altitude and sunshine duration) as covariates, thus greatly improving the interpolation accuracy [34]. The interpolated meteorological data are raster data with a spatial resolution of 1 km. We convert the projection type to the same Albers projection type as the NDVI data.

### 3.1.3. Socioeconomic Data

The socioeconomic data included the number of large livestock and sheep during 1978–2015 at the provincial, city, and county levels. The specific data were obtained from the 1990–2016 Qinghai Provincial Statistical Yearbook, while the data for the Tanggula Mountain Town were obtained from relevant documents [35,36] and the Golmud City People's Government Network (http://www.geermu.gov.cn). Large livestock was calculated as four units of sheep [37].

### 3.2. Specific Research Methods

#### 3.2.1. Linear Regression Method

Using a linear regression analysis can simulate the trend of each pixel over the past 34 years [28]. When the slope is positive, it means that the vegetation index increases over time; in contrast, when the slope is negative, it means that the vegetation index shows a downward trend over time. The significance level test is performed on the obtained slope, and the area that passes the confidence test is considered to have a significant vegetation change trend. The formula is as follows:

$$\theta_{slope} = \frac{n \times \sum_{i=1}^{n} i \times NDVI_i - \sum_{i=1}^{n} i \sum_{i=1}^{n} NDVI_i}{n \times \sum_{i=1}^{n} i^2 - \left(\sum_{i=1}^{n} i\right)^2} \tag{2}$$

where $n$ is the time series length of the simulation ($n = 34$), the variable $i$ is the sequence number of the time series, and $NDVI_i$ is the NDVI value of the $i$-th year.

#### 3.2.2. The Cumulative Departure from the Mean

Abrupt climate change is a kind of discontinuity in the process of meteorological element change, which is usually determined by the cumulative departure curve:

$$C(t) = \sum_{i=1}^{t} \left(X_i - \overline{X}\right) \tag{3}$$

where $C(t)$ is the cumulative departure of meteorological elements, and $X_i$ and $\overline{X}$ are the annual and average values, respectively. If the absolute value of the cumulative departure is the largest, the corresponding $t$ is the mutation year.

#### 3.2.3. Correlation Analysis

The Pearson's correlation coefficient is a parameter used to measure the correlation between two variables, $X$ and $Y$ [38], which is generally represented by $r$:

$$r = \frac{\sum_{i=1}^{n}\left(X_i - \overline{X}\right)\left(Y_i - \overline{Y}\right)}{\sqrt{\sum_{i=1}^{n}\left(X_i - \overline{X}\right)^2}\sqrt{\sum_{i=1}^{n}\left(Y_i - \overline{Y}\right)^2}} \tag{4}$$

where $r$ is the correlation coefficient, $n$ is the total number of samples, $X_i$ is the NDVI value in the $i$-th year, $Y_i$ is the temperature or precipitation in the $i$-th year, $\overline{X}$ is the mean of the variable $X$, and $\overline{Y}$ is the mean of the variable $Y$. We used the Pearson correlation analysis combined with the F-test to analyze the correlativity between the NDVI and meteorological data.

## 4. Results

### 4.1. NDVI Distribution Characteristics

The multiyear average NDVI distribution map of Sanjiangyuan from 1982 to 2015 (Figure 2) can reflect the basic characteristics of vegetation conditions in the area over the past 34 years. The maximum annual average NDVI value in the study area was 0.548, and the average was 0.234. The vegetation condition in the southeast was obviously better than that in the northwest, and this change was closely related to the physical geographical environment. In the southeast of Sanjiangyuan, the lower elevation and better hydrothermal conditions were more suitable for vegetation growth, and the vegetation index was obviously higher, such as in Henan, Jiuzhi, Zeku, Gande, Tongde, Yushu and Nangqian Counties. Among them, Henan County was the area with the highest vegetation

index, with a multiyear average NDVI value of 0.436 (Figure 3). The northwestern area had high altitude, poor hydrothermal conditions and a low vegetation index, including the northern part of Maduo and Qumalai Counties, the northwestern part of Zhiduo County and the whole of Tanggula Mountain Town. Desertification land is mostly distributed in these areas.

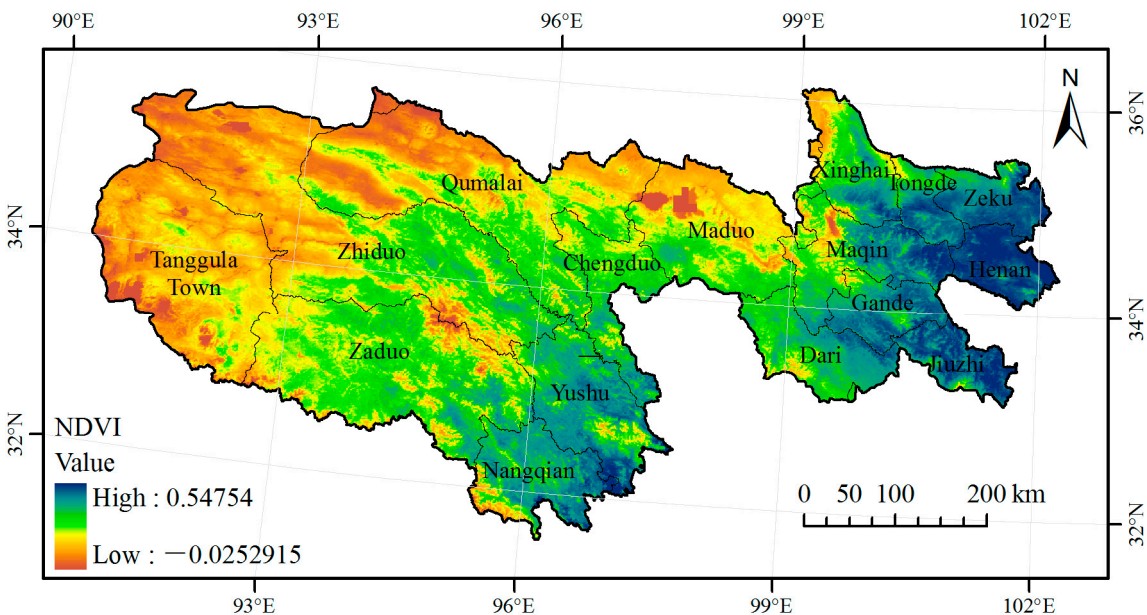

**Figure 2.** Distribution map of the annual average NDVI in Sanjiangyuan over the past 34 years.

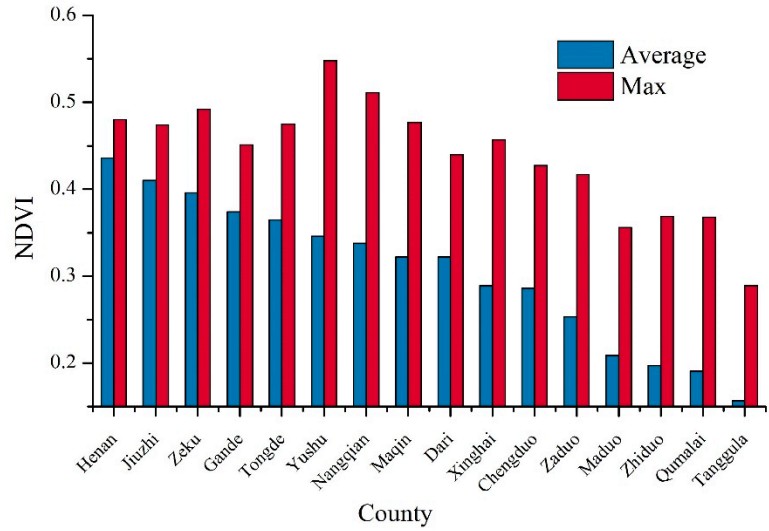

**Figure 3.** The statistics for the average and max annual NDVI in each county.

*4.2. NDVI Interannual Variation Characteristics*

　　To study the characteristics of NDVI interannual variation, the average value of NDVI in the growing season was used to characterize vegetation growth. Vegetation in Qinghai Province generally begins to grow significantly in May and lasts until mid-September. Therefore, May-September is selected as the vegetation growth season in Sanjiangyuan [39,40]. In addition, because there are a large number of non-vegetation areas such as water bodies, glaciers, bare rocks, and sparse alpine vegetation in the study area, it is difficult to obtain the NDVI in areas with sparse vegetation coverage to accurately reflect the vegetation growth status. Therefore, pixels with NDVI less than 0.1 in the following analysis are not included in the statistical analysis [16,19,41].

From 1982 to 2015, the vegetation NDVI in the Sanjiangyuan region showed a fluctuating upward trend (Figure 4), with a growth rate of 0.7%/10a, and it passed the significance test of $p < 0.05$. Dai [42] obtained basically consistent research results by using GIMMS data. The lowest value of the NDVI in the study area was 0.367 in 1995, and the highest value was 0.417 in 2009. The overall trend can be roughly divided into two periods: 1982–2004 and 2005–2015. From 1982 to 2004, the NDVI was low, with a range of 0.367–0.399 and a multiyear average of 0.383. From 2005 to 2015, the NDVI fluctuated greatly and was high, with a range of 0.381–0.417 and a multiyear average of 0.399.

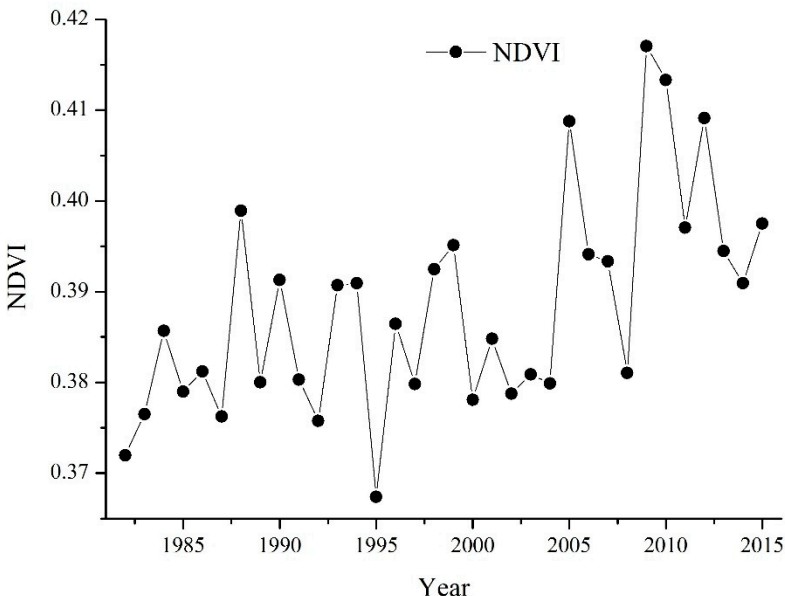

**Figure 4.** Interannual variation trend of NDVI in the growth season of Sanjiangyuan from 1982 to 2015.

### 4.3. The Spatial Variation Tendency of Vegetation from 1982 to 2015

In order to obtain the changing tendency of NDVI, we conducted a pixel-by-pixel linear regression analysis during the 34-year growing season (Figures 4 and 5). We performed a significance test (F test) on the results and divided them into three levels of change: significant (significance level 0.05), weakly significant (significance level 0.1) and insignificant (Table 2). The area of the vegetation NDVI with an increasing trend accounted for 65.6% of the total pixel number, of which the significantly increased area accounted for 53.0%. It was mainly distributed in the eastern part of Sanjiangyuan, including Tongde, Zeku, Henan and Jiuzhi Counties, and other areas near rivers or glaciers. Vegetation in other areas also increased to varying degrees. The area of the vegetation NDVI showed a decreasing trend, accounting for 34.3%, among which the significantly decreased area accounted for 24.0%. The regional distribution of vegetation degradation is scattered mainly in the area near the central part of Sanjiangyuan, including Zado, Yushu, Zhiduo, Qumalai, Maqin and Xinghai Counties.

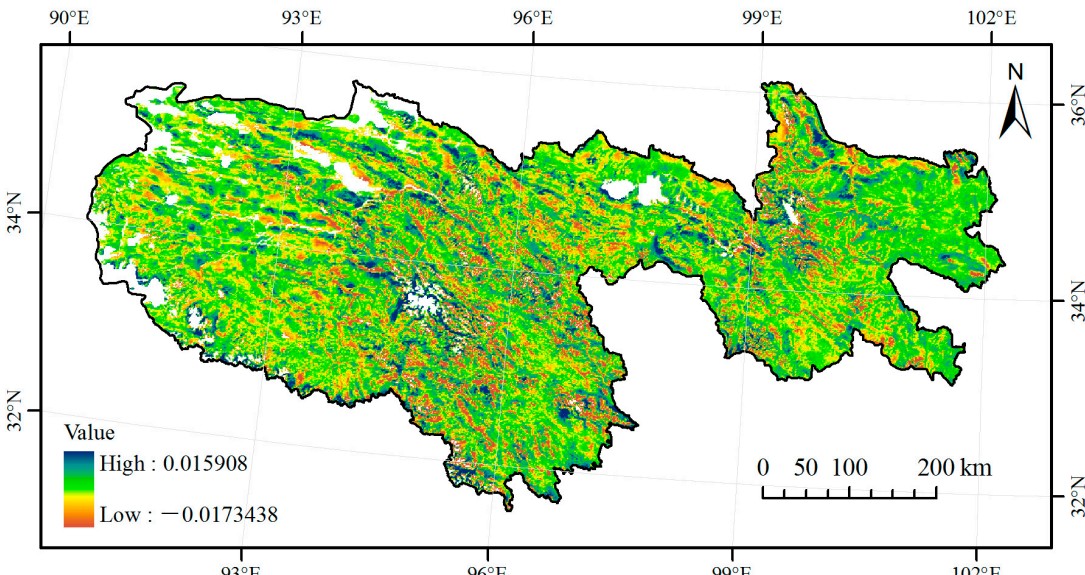

**Figure 5.** Distribution of slopes from the linear regression for the changes in the growing season NDVI from 1982 to 2015.

**Table 2.** Pixel number and ratio of the vegetation NDVI change in the Sanjiangyuan area.

| Change Type | The Significance Level of F Test | Number of Pixels | % |
|---|---|---|---|
| Significant increase | $p < 0.05$ | 138,120 | 53.0 |
| weakly significant increase | $p < 0.1$ | 6008 | 2.3 |
| insignificant increase | $p > 0.1$ | 26,919 | 10.3 |
| Significant decrease | $p < 0.05$ | 62,467 | 24.0 |
| weakly significant decrease | $p < 0.1$ | 4165 | 1.6 |
| insignificant decrease | $p > 0.1$ | 22,803 | 8.8 |

### 4.4. Spatial Variation of Vegetation in Different Periods

According to the specific implementation time of different ecological engineering projects (Table 3), we selected the annual average NDVI value of the past 3 years at the time node to represent the vegetation status during the implementation period [27]. The difference analysis method can reflect the variation and amplitude of the NDVI in a specific period. Therefore, we used this method to calculate the values for different periods, including T2–T1, T3–T2, T4–T1 and T5–T1, to obtain the spatial distribution of vegetation (Figure 6), and the whole process of change in different periods can be observed. We divided the different values into the categories of decreased severely, decreased slightly, unchanged, increased slightly and increased greatly (Table 4).

**Table 3.** Time stage and engineering projects corresponding to different time periods.

| Periods | Time Stage | Engineering Projects | Project Implementation Time Period |
|---|---|---|---|
| T1 | 1987–1989 | Time period before the projects | |
| T2 | 1998–2000 | PFSYR | 1989–2000 |
| T3 | 2003–2005 | Establishing a national nature reserve | |
| T4 | 2010–2012 | EPCQS I | 2000–2012 |
| T5 | 2013–2015 | EPCQS II | From 2013 to present |

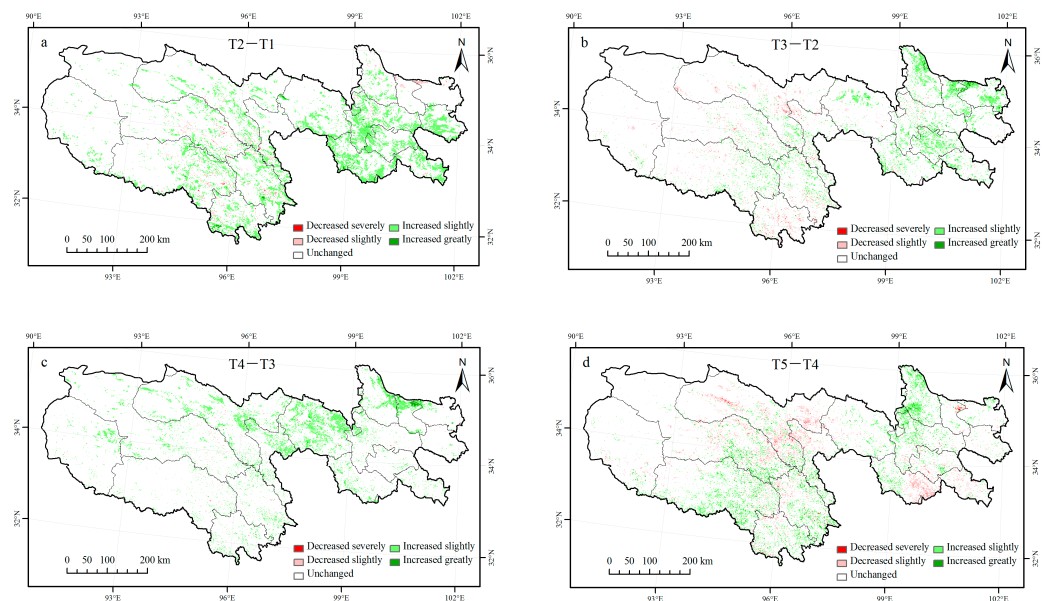

**Figure 6.** The changes in the mean NDVI between different engineering periods: (**a**) change in the mean NDVI from T1 to T2; (**b**) change in the mean NDVI from T2 to T3; (**c**) change in the mean NDVI from T3 to T4 and (**d**) change in the mean NDVI from T4 to T5.

**Table 4.** Analysis of the NDVI values in different periods (in units of pixels).

| Level | | T2–T1 | | T3–T2 | | T4–T3 | | T5–T4 | |
|---|---|---|---|---|---|---|---|---|---|
| Change in NDVI | Level | Number of Pixels | % | Number of Pixels | % | Number of Pixels | % | Number of Pixels | % |
| ≤−0.10 | Decreased severely | 64 | 0.02 | 208 | 0.08 | 36 | 0.01 | 520 | 0.19 |
| −0.10 −0.05 | Decreased slightly | 3709 | 1.35 | 5615 | 2.09 | 927 | 0.34 | 11,341 | 4.21 |
| −0.05 0.05 | Unchanged | 234,808 | 85.24 | 245,416 | 91.30 | 246,885 | 91.34 | 236,769 | 87.93 |
| 0.05 0.10 | Increased slightly | 36,348 | 13.19 | 16,736 | 6.23 | 22,030 | 8.15 | 18,736 | 6.96 |
| ≥0.10 | Increased greatly | 543 | 0.20 | 816 | 0.30 | 412 | 0.15 | 1902 | 0.70 |

The NDVI value of Sanjiangyuan increased significantly compared with T1, especially in the southeast of the region, including in Maqin, Henan, Gande, Dayi and Jiuzhi Counties (Figure 6a). Among them, the areas with slight and significant increases accounted for 13.19% and 0.20%, respectively (Table 4), and the areas with slight and significant decreases accounted for 1.35% and 0.02%, respectively. The areas that did not change were mainly concentrated in the northwest of the study area due to the scarce vegetation and the harsh environment. T2 is the final stage of the PFSYR, which plays a positive role in regional vegetation restoration.

Comparing T2 and T3, the NDVI value increased, but vegetation degradation also occurred in a few areas (Figure 6b). The areas with improved vegetation were mainly distributed in the eastern part of the Sanjiangyuan region, including Xinghai, Tongde, Zeku and Gande Counties, with slight and significant increases accounting for 6.23% and 0.30%, respectively. The areas of decreased vegetation were mainly distributed in Qumalai, Zhiduo, Zaduo and Nangqian Counties, with slight and significant decreases accounting for 2.09% and 0.08%, respectively.

From T3 to T4, vegetation is still increasing, with slight and significant increases accounting for 8.15% and 0.15%, respectively, which are mainly distributed in the central and eastern areas of Sanjiangyuan, including Qumalai, Chengduo, Maduo, Xinghai and Tongde Counties (Figure 6c). These regions are the key remediation areas of the first phase of the EPCQS. The area of vegetation degradation is relatively small, accounting for only 0.35% of the total area, and the significantly reduced area only accounted for 0.01%. T4 is the last stage of the EPCQS I. After 13 years of renovation,

ecological system degradation in Sanjiangyuan has been initially curbed, and vegetation has been effectively restored.

By comparing T5 and T4, it can be seen that vegetation degradation has occurred in some areas (Figure 6d), with slight and significant reductions accounting for 4.21% and 0.19%, respectively, which are mainly distributed in Qumalai, Zhido, Dari and Jiuzhi Counties. However, the increased area of the NDVI is still greater than the decreased area. The areas where the NDVI increases were mainly distributed in Zaduo, Nangqian, Yushu, Xinghai and Maqin Counties, with slight and significant increases accounting for 6.96% and 0.70%, respectively. T5 is the consolidation stage of the EPCQS I and the beginning stage of EPCQS II. During this period, vegetation increased slowly, and vegetation degradation occurred in a few areas.

## 5. Analysis of the Factors Influencing the Vegetation NDVI Changes

### 5.1. Temporal Change Characteristics of Climate Factors

The temporal change in the average annual temperature and its cumulative departure curve from 1982 to 2015 are presented in Figure 6. The average annual temperature exhibited an increasing trend from 1982 to 2015 (Figure 7a). The annual value increased from −1.08 to 1.89 °C, and the multiyear average value was 0.74 °C. This finding is consistent with previous research results [43]. The temperature change point occurred in 1997, and the change could be roughly divided into two periods (i.e., 1982–1997 and 1998–2015) based on the cumulative departure curve (Figure 7b). During the first period (1982–1997), the temperatures remained low, with an average value of 0.15 °C. There were almost negative departures during this period, and the accumulative departure exhibited an obvious decline. The second stage (1998–2015) was a period of relatively high temperatures, and the average value was 1.26 °C. The cumulative departure curve exhibited a steady upward trend during this period, and all cumulative departures in 18 years were positive, indicating a positive change in temperature.

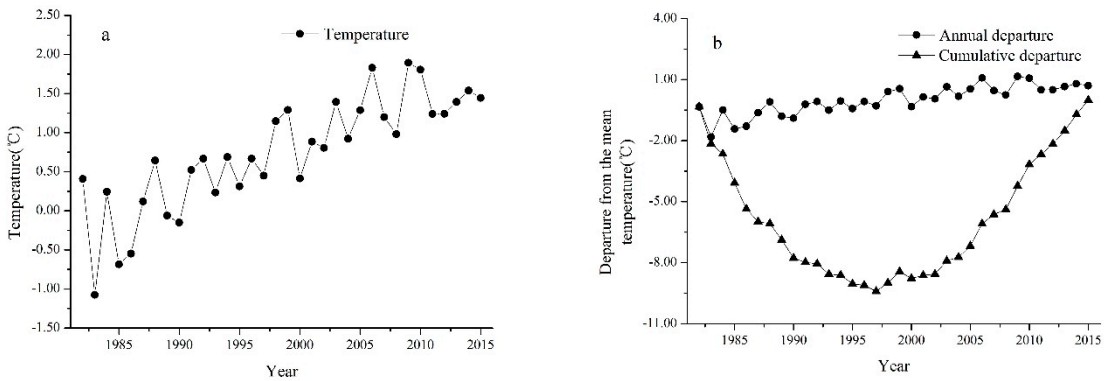

**Figure 7.** Changes in temperatures in 1982–2015: (**a**) annual temperature change and (**b**) changes in annual departure and cumulative departure from the mean temperature.

The temporal change in average annual precipitation and its cumulative departure curve from 1982 to 2015 are presented in Figure 7. The annual precipitation significantly fluctuated within the range of 417.41–609.87 mm, and the average was 511.69 mm (Figure 8a). The precipitation change point occurred in 2002, and this change could be roughly divided into three periods (i.e., 1982–1990, 1991–2002 and 2003–2015) based on the cumulative departure curve (Figure 8b). In the first stage (1982–1990), the fluctuation was severe, and the average annual precipitation was 520.19 mm. The second period (1991–2002) was a period of less precipitation. The annual precipitation varied within the range of 417.41–541.77 mm, with a multiyear average of 480.75 mm. The third stage (2003–2015) was a period of high precipitation, with a multiyear average of 534.37 mm. For the 13 years of precipitation, there was a positive deviation in 9 years and a negative deviation in 3 years, and the cumulative deviation increased significantly, indicating a significant increasing trend for precipitation.

In the three stages, the precipitation in the second stage was at a low level. The average precipitation in the first and third stages was 39.44 mm and 53.61 mm higher than that in the second stage, respectively.

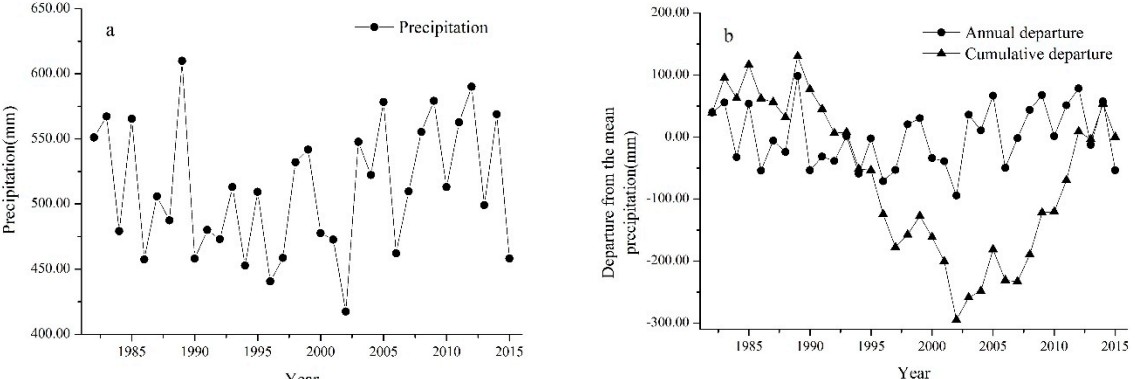

**Figure 8.** Changes in precipitation in 1982–2015: (**a**) annual precipitation change and (**b**) changes in annual departure and cumulative departure from the mean precipitation.

*5.2. Spatial Distribution and Variation Tendency Characteristics of Climatic Factors*

The temperature in Sanjiangyuan shows a gradual decrease from southeast to northwest (Figure 9a). The high-temperature areas are mainly distributed in Tongde, Zeku and Henan Counties in the source area of the Yellow River and in Yushu and Nangqian Counties in the source area of the Lantsang River. The low temperature areas are mainly distributed in the west of Tanggula Mountain Town in the source area of the Yangtze River, the northwest of Zhiduo County and the intersection of Zhiduo, Zaduo and Yushu Counties. The distribution of temperature basically opposite to altitude. The distribution of precipitation shows a trend of gradual decrease from southeast to northwest (Figure 9b). The high-value areas are mainly distributed in Dayi, Jiuzhi and Gande Counties in the southeast of the source area of the Yellow River and in Nangqian and Yushu County of the source area of the Lantsang River. The low-value areas are mainly distributed in the source area of the Yangtze River, covering almost the entire source area, especially in the northwest.

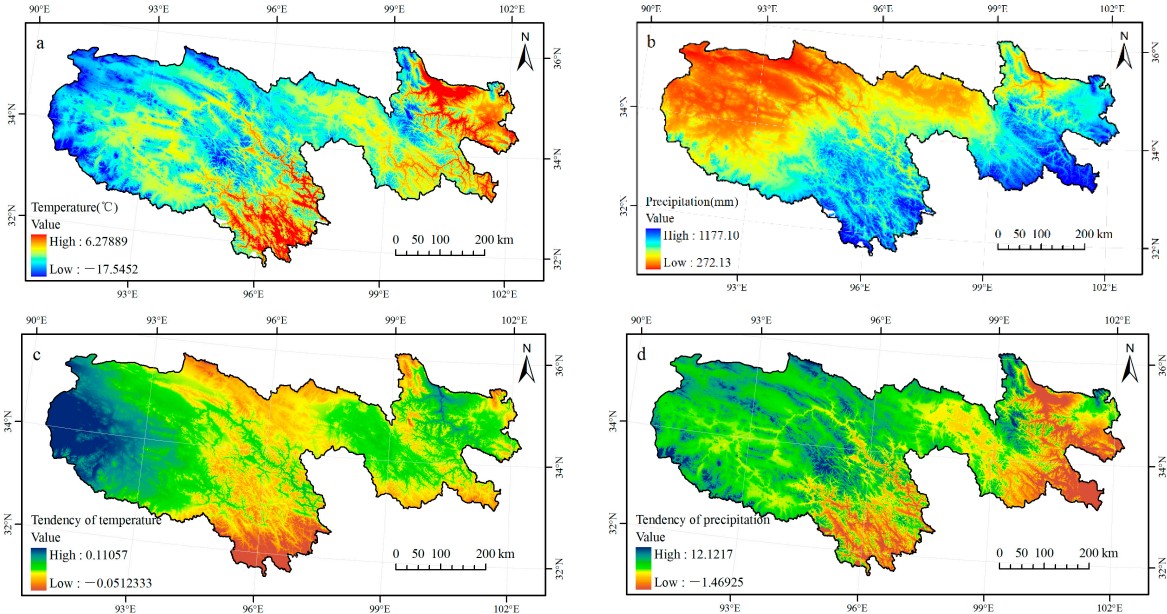

**Figure 9.** (**a**) Multiyear average temperature distribution; (**b**) multiyear average precipitation distribution; (**c**) change tendency of temperature and (**d**) change tendency of precipitation.

We used unary linear regression to calculate the change trend of temperature and precipitation pixel by pixel, and the results were shown in Figure 9c,d. The trend of temperature change showed a state of increasing year by year in the east and west, and decreasing in the middle (Figure 9c). However, the overall variation is not large, and only in the southernmost and westernmost regions, the temperature shows a relatively strong decrease and increase. The trend of precipitation changes showed a decrease in the eastern and southern regions, and an increase in the central and western regions (Figure 9d). Among them, the precipitation in the eastern region decreased significantly, while the precipitation in the whole western region increased significantly.

### 5.3. Correlation Analysis between the NDVI and Climatic Factors

Since the NDVI data from 1982 to 1999 were obtained from the 8 km GIMMS NDVI data through resampling and correlation calculations, the pixel distribution had a smooth gradation. When the correlation between the NDVI and climate factors was calculated pixel by pixel, the data caused a large error. To better express the correlation between climate factors and the NDVI, we used the MODIS NDVI data from 2000 to 2015 for calculations and analyses.

The correlation coefficients between temperature, precipitation and the NDVI from 2000 to 2015 were calculated on a pixel-by-pixel basis (Figure 10). The NDVI of Sanjiangyuan is basically positively correlated with temperature (Figure 10a). The number of positively correlated pixels accounted for 89.7%, and the number of negatively correlated pixels accounted for 10.3%. Among them, 16.5% and 8.6% of the area passed the $p < 0.05$ and $p < 0.01$ significance tests, respectively. The regions with high correlation are located in the west and southeast, including in Tanggula Mountain Town, Zhiduo County and the northwestern part of Qumalai County in the source area of the Yangtze River and Maduo, Dari, Gande, Zeku and Henan Counties in the source area of the Yellow River. There was also a basically positive correlation between the NDVI and precipitation (Figure 10b), with 84.3% of the positively correlated pixels and 15.7% of the negatively correlated pixels. Among them, 16.4% and 7.9% of the areas passed the $p < 0.05$ and $p < 0.01$ significance tests, respectively. The regions with high correlation were distributed in Qumalai County and at the intersection of Tanggula Mountain Town and Zhiduo and Zaduo Counties in the source area of the Yangtze River, and all counties in the source area of the Yellow River except Jiuzhi, Henan and Maqin Counties.

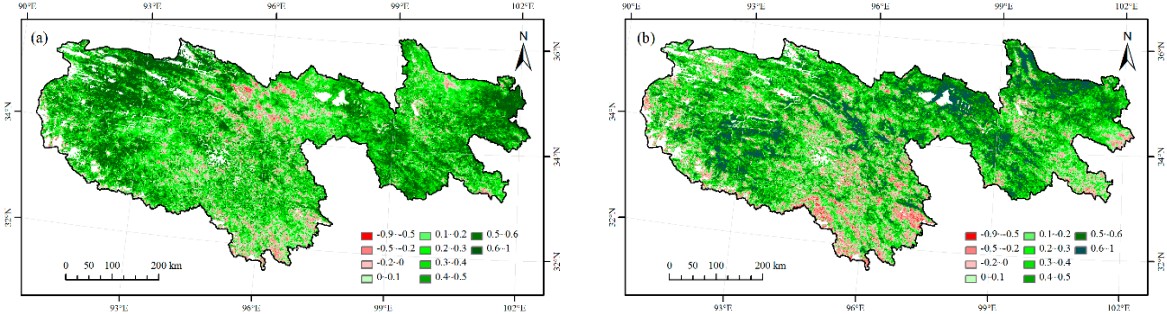

**Figure 10.** (**a**) The spatial distribution of the NDVI and temperature correlations and (**b**) the spatial distribution of the NDVI and precipitation correlations.

The main climatic factors that affect vegetation growth in different regions of Sanjiangyuan are different. In the northeastern part of the study area, vegetation growth is mainly affected by precipitation, while in the western region, it is more obviously affected by temperature, while in the southern part, it is not obviously affected by either temperature or precipitation.

On a source region scale, the vegetation growth in the source area of the Yangtze River was affected by precipitation as a whole, while the western region was more affected by temperature; the northern part of the source area of the Yellow River was significantly affected by precipitation,

while the southern part was more affected by temperature, while the source area of the Lantsang River was not significantly affected by the two climatic factors.

Generally, water is the main factor determining plant growth in the caloric surplus area. In the heat deficit area, the hydrothermal condition of soil is the main factor of plant growth. In the area of severe heat shortage, temperature is the main factor restricting plant growth. In the source area of the Yangtze River, the overall elevation is high, and the temperature and precipitation are both low. For example, at Wudaoliang and Tuotuohe Stations in the west of the source area, the average temperature from 1982 to 2015 was as low as −5.49 °C and −3.51 °C, respectively (Table 5), and the annual precipitation was 297.67 and 307.69 mm, respectively. Plant growth is adverse to continuous low temperatures, which reduces photosynthesis, affects the absorption of minerals and the operation of nutrients, and inhibits the metabolism of roots, stems and leaves; the result is slow growth and development. Therefore, in the western low temperature region, the vegetation NDVI is closely related to temperature.

**Table 5.** Multiyear average temperature and precipitation at different weather stations from 1982 to 2015.

| Weather Stations | Wudaoliang | Tuotuohe | Maduo | Dari | Tongde | Jiuzhi | Nangqian |
|---|---|---|---|---|---|---|---|
| Temperature (°C) | −5.49 | −3.51 | −3.24 | −0.59 | 2.48 | 1.04 | 4.51 |
| Precipitation (mm) | 297.67 | 307.96 | 362.58 | 567.25 | 421.19 | 685.57 | 517.30 |

In the source area of the Yellow River, the altitude is lower, and the hydrothermal conditions are better. Therefore, the vegetation in different regions is affected to various degrees by temperature and precipitation. For example, the annual average temperatures at Maduo and Dari Stations are −3.24 °C and −0.59 °C, and the precipitation is 362.58 mm and 567.25 mm, respectively. This area is a heat-deficient area, and its vegetation is significantly affected by temperature and precipitation. Maduo County is more affected by precipitation due to less precipitation. The Tongde Station in the northeast has high temperatures and low precipitation areas, so the regions with similar natural conditions are mainly affected by precipitation, such as Xinghai, Tongde and Zeku Counties. In the southeast, where hydrothermal conditions are good, such as at Jiuzhi Station, vegetation is not significantly affected by the two factors. Similarly, in the source area of the Lantsang River, the hydrothermal conditions are good, such as at Nangqian Station, temperature and precipitation are no longer the main limiting factors for vegetation growth, and the relationship is not obvious.

*5.4. Human Activities*

Although climate change is an important factor in the vegetation cover change in the Sanjiangyuan area, human activity is also a driving factor that cannot be ignored. Generally, factors affecting regional vegetation coverage include agricultural production, industrial production, urban construction and ecological construction projects. In Sanjiangyuan, there are few agricultural areas, industries and towns, so we only discussed the impact of animal husbandry and ecological construction projects on vegetation coverage.

Animal husbandry is the main industry in the Sanjiangyuan area. Before 1995, the utilization of grassland was mainly based on extensive management, with a large number of livestock (Figure 11), high pressure on grassland stocking and a serious plunder of grassland resources. According to research [44], in the 15 years before the implementation of the livestock reduction project (1988–2002), the grassland was overloaded by approximately 1.29 times. Serious overload led to grassland quality degradation and serious vegetation degradation, and the vegetation NDVI remained low during this period. In 2003, the livestock reduction project was implemented, and the grassland utilization mode was changed from extensive to intensive. The construction of livestock breeding projects such as livestock greenhouses, hay storage sheds, and artificial feed planting bases were used to improve the quality of livestock and optimize the structure of the herd. From 2003 to 2012, the grassland was overloaded by approximately 0.46 times, the livestock pressure was greatly reduced, and the forage

yield was significantly increased [44]. Vegetation restoration was accelerating, and the vegetation NDVI was increasing during this period.

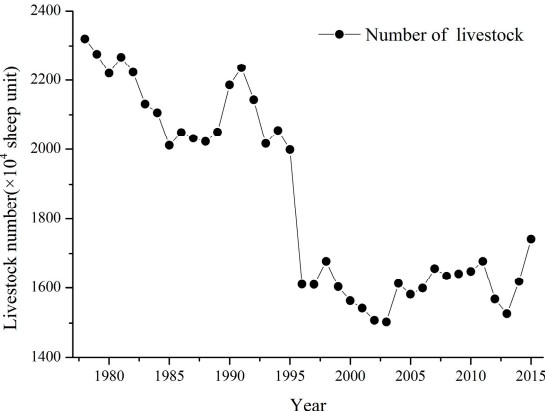

**Figure 11.** Changes in the number of livestock from 1978 to 2015.

In addition to the construction of livestock projects, the government has implemented a number of ecological construction policies and measures in Sanjiangyuan. In 1989, the state approved the first phase of the PFSYR for 145 counties in nine provinces, including Qinghai Province. On the basis of protecting the existing vegetation, the project adopts various forms to build new vegetation groups with unified benefits such as shelterbelt, timber forest and economic forest. By 1997, the forest coverage rate in Qinghai Province had increased to 3.11%, and the regional vegetation ecosystem had been effectively restored [14]. In 2003, the government established a national nature reserve in Sanjiangyuan and carried out the first phase of ecological protection and construction in 2005. EPCQS I plans to protect an area of 152,300 km$^2$, with an investment of 7.5 billion yuan. The main ecological measures included the following aspects: (1) Returning the grazing land to grasslands, and determining the number of livestock by grass: In areas with severe desertification, fencing and grazing prohibition were implemented, and in areas with less desertification, the number of livestock was determined by the grass. These measures could control the number of livestock, and gradually restore grassland productivity. (2) The livestock reduction project: This was implemented to improve the quality of livestock and optimize the structure of the herd. The construction of livestock raising projects were carried out, such as building livestock greenhouses and grass storage sheds, establishing artificial feed planting bases, and so on. (3) Ecological migration: The herdsmen were removed from meadows where desertification was serious, and engaged in meat and wool product processing, in order to reduce the pressure on grassland carrying livestock. (4) Returning farmlands to forests or grasslands: In areas with more cultivated land, this policy was implemented to alleviate the ADG caused by farmland reclamation. (5) Artificial precipitation: Artificial precipitation was adopted to increase the regional precipitation, promote vegetation restoration and curb the process of desertification. Studies [9,45] have shown that at the end of the EPCQS I, the ecosystem structure has been partially improved, the grassland degradation trend has been initially curbed, the ecological restoration of severely degraded areas is obvious, the productivity of grassland has increased and the pressure on grasslands that are used for animal husbandry has been effectively reduced. However, the improvement trend of nature reserves and key engineering areas is obviously better than that of surrounding areas.

## 6. Discussion

In this study, the NDVI value had a strong upward trend from 1982 to 2015. Previous studies have also confirmed that the vegetation conditions of Sanjiangyuan have improved in the past few decades [10,11]. These results show that under the comprehensive influence of various ecological projects and natural factors, the vegetation of Sanjiangyuan has been restored. Based on the analysis of vegetation cover changes of major ecological environmental projects in the past 34 years, the results

show that ecological restoration plans, such as the PFSYR and the EPCQS I and II, are effective for vegetation restoration in the Sanjiangyuan region.

Some studies have pointed out [10] that climate warming is a decisive factor affecting vegetation growth in the Sanjiangyuan region. Since 2000, except for the aridity in the western part of Sanjiangyuan, most of the other regions have a warm and humid trend [46]. This trend determines the growth trend of vegetation in the Sanjiangyuan region. Many studies [47,48] have suggested that compared with the increase in precipitation, the increase in temperature was the main factor in vegetation growth. In this paper, it is believed that the influence of temperature and precipitation on vegetation growth in different regions varies due to the great differences in elevation and climate conditions in different regions of the Sanjiangyuan region. However, from the overall regional perspective, the pixel number of the positive correlation between vegetation and temperature is slightly larger than that between vegetation and precipitation, which also indicates that temperature plays a more important role in vegetation change in the Sanjiangyuan region.

The impact of human activities on the vegetation changes in Sanjiangyuan mainly includes animal husbandry and ecological engineering, which have both negative and positive effects. With the continuous increase in the population, the negative impact of human activities on the ecological environment is also increasing, such as overgrazing caused by the rapid increase of livestock [49], the destruction of surface vegetation caused by sand mining [50], the decline of soil fertility caused by large-scale burning of livestock manure [51,52] and the impact of *Cordyceps* excavation on vegetation coverage [53]. However, human beings also improve regional ecological environmental quality through the rational use of resources, such as reducing environmental pollution, protecting forests, improving energy structure, and implementing a series of large-scale green projects [54,55]. Since 1989, the government has implemented a series of ecological environmental protection measures in the Sanjiangyuan area. After decades of efforts, these measures have played an important role in restoring the ecological environment and slowing the trend of land desertification and grassland degradation. Studies [10,48] have shown that from 2000 to 2010, human activities had a positive impact on the vegetation cover changes in the Sanjiangyuan region.

This article made several improvements on the basis of previous studies. Firstly, instead of using a single data source, GIMMS NDVI and MODIS NDVI were combined to solve the problem of low resolution of the former and short time period of the latter. Additionally, spatial and temporal change analysis of vegetation is carried out for as long as 34 years. Secondly, we discussed the vegetation changes based on the time period of the ecological restoration project, and tried to find how the vegetation changes during the different ecological engineering, where the changing areas are, and how the vegetation increases or decreases. Thirdly, we explored the spatial trends of climatic factors, which helped us to have a deeper understanding of the impact of climate on regional vegetation. Although the resolution of the NDVI data used in this article was higher than that of other long-term NDVI products, the resolution of 1 km was still insufficient for the Sanjiangyuan area. In some complex terrains, slope and other topographic elements had a certain effect on vegetation reflectance, which in turn affected the size of the NDVI. In addition, sparse vegetation will also affect the reflectivity in arid areas, such as the northwestern part of the study area. In addition, to reduce the error in the calculation of correlation between climate factors and vegetation, only the data after the year 2000 were used. The next step should be to find a better way to eliminate this error.

## 7. Conclusions

In this study, we used the GIMMS NDVI and MODIS NDVI data to monitor the vegetation change of the Sanjiangyuan area under the influence of ecological engineering from 1982 to 2015 and analyzed the driving factors behind it. The results show that during the past 34 years, vegetation coverage in the study area was increasing continuously, which was influenced by both climate change and human activities. In different regions, the hydrothermal conditions were different, so the climatic factors affecting vegetation change were not the same. In different time periods, human activities have had

different impacts on vegetation. The rapid development of animal husbandry in the early stage has had negative impacts on vegetation, while a series of ecological projects implemented in the later stage have had positive impacts on vegetation growth.

There are still some limitations of this research, but we used 1 km-resolution data to monitor vegetation dynamics over a long period of time, discussed changes in vegetation at different stages of the ecological project and revealed the main climatic factors that affect vegetation changes, thus providing a scientific basis for the evaluation of regional ecological engineering and the formulation of ecological protection policies.

**Author Contributions:** Conceptualization, X.Z.; Data curation, X.Z., X.X. and X.W.; Formal analysis, X.Z.; Funding acquisition, X.L. and C.Y.; Investigation, X.Z.; Methodology, X.Z. and X.L.; Software, X.Z., H.J. and K.F.; Supervision, C.Y.; Validation, C.Y.; Writing—original draft, X.Z.; Writing—review & editing, X.L. All authors have read and agreed to the published version of the manuscript.

**Funding:** This work was funded by the Natural Science Foundation of China (Grant No. 41730752 and 41971277).

**Conflicts of Interest:** The authors declare no conflict of interest.

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
