# Peer review of "Vegetation Dynamic Changes and Their Response to Ecological Engineering in the Sanjiangyuan Region of China"

_remotesensing, doi:10.3390/rs12244035_

Round 1

Reviewer 1 Report

In my opinion, the article submitted has conditions to be published. The paper is interesting and presents an updated, valuable, and extensive state of art.

The Discussion and Conclusions chapter integrates the results well and alerts to the challenges between climate change and human activities

Author Response

Thank you!

Reviewer 2 Report

General Comments

Liked the paper. Well written and clear. The main observation is that climate change (increased rainfall and temperatures-- especially in the second stage) appears to best explain increased NDVI over time.

It's harder to tease out the effects of policy. In fact, the dominant category for "Analysis of the NDVI values in different periods (in units of pixels)", Table 5 is "Unchanged," ranging between 85% to 91% across time periods. "Increased slightly" is the next category, which bodes well for LULC.

Therefore the main conclusion in Lines 398-399 “The results show that under the comprehensive influence of various ecological projects and natural factors, the vegetation of Sanjiangyuan has been restored.” Thus policy implementations appear to help, but insufficient information is provided, such as if these policies were coercive. What incentive/ disincentive structures were provided? Where did the livestock go in the livestock reduction program? What were reforestation programs like? Did they use autochthonous vegetation/trees, or fast growing (and potentially thirsty) exotics? Findings might be overly optimistic regarding policy interventions. Perhaps I’m wrong. 

————————

Specific Comments

Liked the paper and the way it was organized. Convincing justification for pursuing this topic was provided.

Important that authors included climatological data (rainfall and temp) to help explain observed NDVI changes.

The use of various maps throughout the paper is very useful (consider using more if possible). 

Isn’t resampling 8km to 1km data problematic, such as was done with GIMMS NDVI?  Also, paper states that these data have “smaller errors and higher accuracy…” (Line 92). This may be so, but is surprising at 8km spatial resolution, meaning 1 pixel = 8,000 meters on the ground. Challenges with this low spatial resolution are mentioned in Section 4.3, Lines 300-304, where correlating NDVI to climate factors by pixel becomes problematic. 

Agree with the use of MVC (maximum value composite) for each month.

Also agree with the method of excluding NDVI pixels with a value less than 0.1, so as to exclude sparse vegetation, glaciers, bare rocks and water bodies, as the authors note (Lines 174-175). 

Aside from maps, inter annual NDVI trend charts, such as Figure 3, are useful. 

A bit unclear, or perhaps I misunderstood, but Lines 207-208 say: “we selected the annual average NDVI value of the past 3 years at the time node to represent the vegetation status during the implementation period.” Does this mean that 3 years after policies were implemented, a lag was provided to test the policy effects?”

Important, but perhaps expected finding: “The distribution of temperature basically conforms to the distribution of altitude.” (Lines 289-290). Perhaps also expected is the converse, lower altitude, higher temperature, and in this case, also wetter. We would additionally expect these areas to be more populated and thus have higher anthropogenic pressures. Thus (Lines 322-325) the authors find that vegetation growth in the western (higher altitude, colder region) is mainly affected by temperature, while in the northeastern part (warmer region) it “is mainly affected by precipitation.” Interestingly, the southern part “is not obviously affected by either temperature or precipitation,” which appears to be a normally warmer and wetter part of the study area.

Lines 371-372 mention livestock grazing was changed from extensive to intensive. Perhaps provide additional information. Plots changed from X hectares to Y hectares? What was the land tenure regime on these plots? 

It appears that reducing grazing pressures indeed worked in increasing vegetation growth (Lines 376-377).

While rainfall and temperature tend to dictate vegetation growth and abundance, the authors found that the influence of temperature on vegetation growth was slightly larger than that of precipitation. Important finding, particularly as global temperatures increase. In congress with these observations are those by other scientists who point out that France’s great wine growing regions will eventually move to England. 

A little more effort could be spent addressing the limitations of the study (Lines 428-434). Here it is apparent that data used to test the correlation between climate and vegetation only goes back to 2000. And while that are the data the authors had to work with, when it comes to climatological-related findings, 20 years may be insufficient to draw strong conclusions. 

Lastly, the authors mention the use of “high-resolution data to monitor vegetation dynamics…” (Line 447). Neither 1km, and especially 8km spatial resolution data are NOT considered high resolution. While this may be what’s available for NDVI generated information, and perhaps there’s a different classification here, high-resolution imagery would  like include 5m and below. 

Author Response

Thank you!

Reviewer 3 Report

Introduction
The introduction is not well structured, I think that the initial part, from line 31 to line 50 can be reformulated and may be moved in the study areas.
The Authors, need to report the state of the art and how it improves that

Methods
The work has a good aim and the authors use a good approach to carried out it.
but I didn't understand if the methodology used for data correlation was improved by the authors? Maybe it is better to explain the innovation of the methodology
I will suggest focusing on the approach (combination of different data and use different time window for the analysis) more than on satellite data fusion that part of the approach (of course, this is my suggestion). therefore, you can start the section with a part that explains the general approach and after that the use of remote sensing data correlation how a tool to apply your approach.

Conclusion
some parts are similar to the discussion therefore, I find the same information. Maybe you can improve it considering that in your work, it is not important only the technic of the remote sensing used, but also like it is applied. in this case, I think that the technic was calibrated considering the aim of the work. and in this case, I can choose to accept the low resolution. because considering the aims, this is the only solution.

Author Response

Thank you!

Reviewer 4 Report

The manuscript "Vegetation Dynamic Changes and their Response to Ecological Engineering in the Sanjiangyuan Region of China" has merit to be published in Remote Sensing. However, it is necessary to go through Major Revision before being accepted. Below, I send comments to the authors:

- define the GIMMS NDVI and MODIS NDVI indexes in the Abstract. Also, define abbreviations throughout the text when you quote them for the first time;
- the presentation of the results in the Abstract is not good. The authors need to conclude by demonstrating the main applications of this study;
- I suggest that the authors make it clear what gap can be filled with the proposed study. This needs to be done in a paragraph at the end of the introduction (before the objectives);
- include your research hypotheses;
- what is the scale (high and low) shown in Figure 1? I suppose it is from precipitation, but this needs to be made clear in the caption of this Figure;
- It would be interesting to apply the Man-Kendall test to see if there is a tendency to increase or decrease the NDVI and precipitation values. This analysis needs to be done over the years and it would be important to understand the behavior of the data;
- Replace the data in Table 2 with a boxplot for each variable and region. This will allow you to better see the variation in the data;
- what does the p-value shown in Table 3 indicate?
- In general, the figures are of low quality. Figures 6 "a" and "b" could be joined. The same is true for Figures 7 "a" and "b";
- Study discussion is poor. It does not demonstrate the progress of this manuscript in relation to those already published. This needs to improve a lot.

Author Response

Thank you!

Round 2

Reviewer 3 Report

I appreciated the changes made. I believe the work deserves to be published

Reviewer 4 Report

The authors made the necessary corrections and the manuscript can be accepted in this way.